computer vision/ecology/computational biology

individual re-identification, photo identification, deep-learning, spot extraction, spot matching, capture–mark–recapture

**Author for correspondence:**
Elizabeth A. Mittell
e-mails: em294@st-andrews.ac.uk, e.mittell@gmail.com

# Re-identification of individuals from images using spot constellations: a case study in Arctic charr (*Salvelinus alpinus*)

Ignacy T. Dębicki[1], Elizabeth A. Mittell[2,3,†],
Bjarni K. Kristjánsson[3], Camille A. Leblanc[3],
Michael B. Morrissey[2,‡] and Kasim Terzić[1,‡]

[1]School of Computer Science, and [2]School of Biology, University of St Andrews, St Andrews, UK
[3]Department of Aquaculture and Fish Biology, Hólar University, Sauðárkrókur, Iceland

EAM, 0000-0002-5801-614X

The ability to re-identify individuals is fundamental to the individual-based studies that are required to estimate many important ecological and evolutionary parameters in wild populations. Traditional methods of marking individuals and tracking them through time can be invasive and imperfect, which can affect these estimates and create uncertainties for population management. Here we present a photographic re-identification method that uses spot constellations in images to match specimens through time. Photographs of Arctic charr (*Salvelinus alpinus*) were used as a case study. Classical computer vision techniques were compared with new deep-learning techniques for masks and spot extraction. We found that a U-Net approach trained on a small set of human-annotated photographs performed substantially better than a baseline feature engineering approach. For matching the spot constellations, two algorithms were adapted, and, depending on whether a fully or semi-automated set-up is preferred, we show how either one or a combination of these algorithms can be implemented. Within our case study, our pipeline both successfully identified unmarked individuals from photographs alone and re-identified individuals that had lost tags, resulting in an approximately 4% increase in our estimate of survival rate. Overall, our multi-step pipeline involves little human supervision and could be applied to many organisms.

[†]These authors contributed equally to this study.
[‡]Joint senior authors.

# 1. Introduction

Individual-based studies of animal populations support investigations of a wide array of ecological and evolutionary topics, often facilitating some of the most powerful and robust analyses, based on the longitudinal data they generate [1]. A fundamental element of any individual-based study is a mechanism to recognize individuals, with a wide range of approaches employed across studies [2]. These range from visual recognition by experienced fieldworkers (*in situ* or from videos and photographs; [3]), to the use of visually read identifier tags, electronically read tags, active telemetry (e.g. for obtaining individual bird flight paths; [4]) and genetics [5]. Each of these methods comes with different challenges. For example, visual identification typically requires extensive system-specific expertise, some types of tags are costly, may be lost, excreted or become non-functional, and it can be difficult and expensive to obtain genetic samples. Additionally, all methods of identification, including visual identification without the aid of non-natural markings, can potentially cause changes (e.g. behavioural or energetic) in animal subjects. Re-identification of animals from photographs has highly desirable features for many systems, such as potentially minimizing intrusiveness, and may often represent a low-cost and practical means of supplementing other means of identification. However, the number of comparisons among photographs required to make identifications can be very large, the availability of features on which to verify identities may be unclear, and the technical capacity for such features to be captured and interpreted on photographs may be difficult to develop. The potential for photographic identification of wild animals for ecological studies will therefore require advances in computational methods to support several steps of the process before its potential benefits can be widely realized.

Physical markers or tags are frequently used to keep track of individuals over time and obtain individual-based data. Older, cheaper methods rely on external tags or marks, such as anchor tags and nitrogen branding, whereas newer methods use small transponders injected under the skin, such as passive integrated transponder (PIT) tags [6,7]. However, even with newer tagging methods, the effects of the tagging on a study can be non-trivial as the behaviour and survival of the organism can be affected and, notably, tag loss can occur—increasing margins of error [6,8]. More recently, natural markings on fish and other animals have been used instead of artificially placed marks to identify individuals using photos. This 'photo-ID' system has been trialled on whale flukes [9], but with only moderate success due to the very limited range of features humans can easily identify and match, especially from older photos [10]. More recent attempts have been more successful, using the vastly increased computational capabilities of modern computers, to match cheetahs [11], elephants [12], penguins [13], sharks [14] and chimpanzees [15] among others. Using photo-ID by itself can make studies less invasive to the study organism, which is important for studies of wild populations. Alternatively, these methods can be used to reduce the effect of tag loss on studies, without significantly increasing processing time when having to handle large numbers of individuals.

For photo-ID to be feasible in large studies, computer automation tools are required. This is because the number of comparisons quickly grows as the database size increases. So far, multiple computer tools have been developed to help speed up the process, from the original by-eye matching to scalable high-throughput solutions for managing large databases of images [16]. One of the first, and still most common, elements of photo-ID to be automated is the matching, where features are annotated by hand, but the matching of individuals is automated [10]. These systems are 'feature engineered', where the features used for matching are devised by a human for that purpose. For example, the spot matching systems developed by Arzoumanian *et al.* [17] and Van Tienhoven *et al.* [18], use spots to match individuals, and for each image, the user manually inputs the location of spots and reference points. This greatly improves the overall processing time, as the human time spent on the study now grows linearly with the increasing number of individuals (e.g. $n$ min), rather than with the square of the sample size (e.g. $n^2$ min). However, with larger numbers of samples, such as those generated by capture–mark–recapture (CMR) studies, this can still be very intensive, with manual marking and human verification of matches taking over a minute per image [19]. Furthermore, manual marking can be more error prone when multiple observers perform the marking and when there is variation in what is considered to be a spot between each person [20]. This can result in large discrepancies, for example, a difference of 30% in the accuracy rate of some matching algorithms [19], and emphasizes the need to improve photo-ID automation tools.

In order to overcome some of the time constraints, software has been developed to automate the feature extraction process, such as the automated fin extraction used by 'Finscan' [21] or the manta ray matcher that uses the scale-invariant feature transformation (SIFT) algorithm [22,23]. However, these systems are not widespread due to their highly specific application, and the rapid increase in

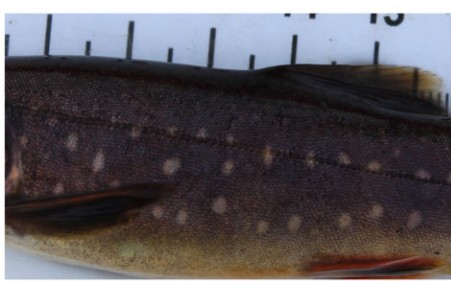 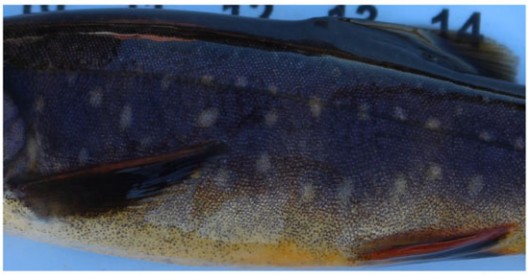

**Figure 1.** Examples of spot patterns on the flank of a small benthic Arctic charr *Salvelinus alpinus*, taken 1 year apart.

use of neural networks and deep learning within computer science [10]. These newer systems use deep learning to automate the entire process, using convolutional neural networks (CNNs) to automate both feature selection and detection, and promised to be more generalizable across more species [24]. A downside of using deep learning for automated feature selection and extraction is that it requires a large number of photos to properly train the network, which may not be available at the start of a study for a wide variety of species [10]. Furthermore, a common feature of many animals is spots, and despite rapid progress, fully automated publicly available systems for photo-ID using spots are rare. The Interactive Identification System ($I^3S$; [18]) is the most feature complete (available to use but still containing bugs); however, because its automated feature extraction relies on key-point extraction and SIFT matching, it is susceptible to poor-quality photos and performs poorly on regularly sized spots, which are not highly distinctive on their own, but are arranged in distinctive patterns. There is therefore a gap in the public domain for automated feature extraction, particularly focusing on spot patterning.

Small benthic Arctic charr (*Salvelinus alpinus*) are found in spring-fed systems within the volcanically active area of Iceland [25] and exhibit natural spot markings on their flanks (figure 1). We use this study system to describe and develop a multi-step fully automated pipeline for matching specimens based on their unique spot markings. As part of the development of this pipeline, we compared both traditional computer vision techniques and new deep-learning techniques to identify and extract detailed fish masks and spot annotations from raw images. Finally, we explore two spot matching algorithms: an extension on the Groth matcher, initially developed by Groth [26] for identifying star constellations, before being adapted by Arzoumanian *et al*. [17] for spot matching in whale sharks; and a random sample consensus (RANSAC)-based algorithm, also initially developed for star constellation matching by Beroiz *et al*. [27], which we extend to make additional use of fish masks to help limit the search space when a large number of spots are present. In our case study on Arctic charr, the spots are generally consistent in size and colour, so improvement on current algorithms in this system with these generic spots should make these algorithms more widely applicable. In addition, we use our method of re-identification to correct fish capture histories in the Arctic charr system and estimate survival rate based on uncorrected and corrected capture histories. This demonstrates that the more precise knowledge of individuals within a population that our method allows can have an impact on biological properties important for population management, ecological and evolutionary studies.

# 2. Methods

## 2.1. Study system

Since 2012 a CMR study has been running on Arctic charr populations inhabiting 20 small lava caves in the vicinity of Lake Mývatn in northeast Iceland. Each year the populations are sampled twice; at the beginning of the growing season in June and during the growing season in August. At each sampling occasion, fish are captured by a combination of electrofishing and non-baited trapping. All unknown fish that are captured are PIT tagged, measured for fork length (mm), fin clipped (as a genetic sample) and photographed on their left side. An exception to this data collection scheme was in the first capture occasion (June 2012) when fish were caught, photographed and genetic samples were taken but no tagging was carried out. In the course of data collection, it has become apparent that some fish have lost their PIT tags, i.e. an absence of a tag despite evidence of previous surgical implantation and/or genetic sampling. Loss of PIT tags is known to occur in a range of fish species, which is often true if they were implanted recently [8]. Preliminary informal studies within this Arctic

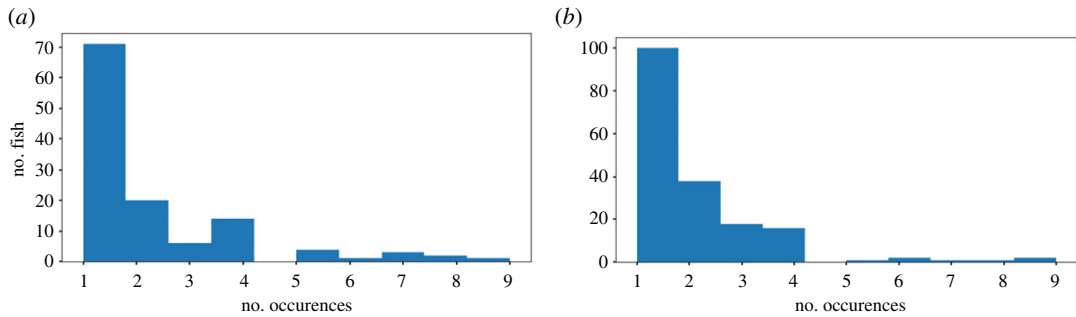

**Figure 2.** Histograms (*a*) and (*b*) show the frequency of (re)captures for individuals identified in Caves A and B, respectively.

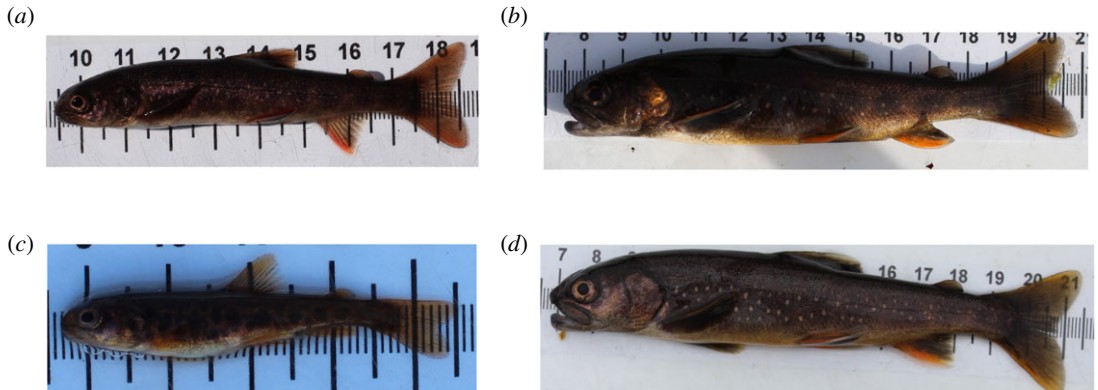

**Figure 3.** Examples of the variation in quality of photographs of fish in the case study system. (*a*) A poor quality image of a fish with most spots not visible. (*b*) A fish partially in shadow. (*c*) A young fish with parr marks rather than spots. (*d*) A good quality image of fish.

charr system have suggested that older individuals appear to lose their tags more often than younger individuals (CA Leblanc 2019, personal observation). Re-identification of individuals from photographs by an expert is possible using the spot patterning in this system. However, the process is extremely time-consuming and the number of photographs is too great to correct all the errors in recapture histories arising from PIT tag loss. It is currently unknown at what rate tag loss may occur in the Arctic charr inhabiting lava caves around Lake Mývatn, although field observation suggests that losses occur at a low but non-trivial rate.

To develop a strategy for photographic identification in these Arctic charr, we focused on a dataset of 261 images collected between 2012 and 2017, including 135 known recaptures from 124 unique fish (as discernible from retained PIT tags), from one cave (hereafter 'Cave A') for development and testing of methods (the distribution of recaptures per individual is shown in figure 2*a*). To independently verify the performance of the methods, we used a second dataset of 346 images also collected between 2012 and 2017, but from a different cave (hereafter 'Cave B'). This second dataset included 165 resolved recaptures of 160 known individuals (figure 2*b*). In these data, using previous methods of re-identification, most individuals are only seen once or twice. All photographs are accompanied by individual identification information, as far as it is provided by retained PIT tags and fork lengths (mm), and all photographs are taken on a light background with a ruler in the image for scale.

The photos used in tool development were not filtered for quality, so they represent a real-world situation (i.e. changing conditions during the processing of animals on site). Some examples of good-, medium- and poor-quality photos are shown in figure 3. In order to allow the pipeline to be evaluated without any training bias, Cave A was used for training, whereas Cave B was only used as a testing set to independently verify methods. Of the 261 images in Cave A, 52 were selected at random and annotated for the fish location within the image (fish masks) and the spots on the fish. The fish masks and spots were annotated using black and white masks in the photo editing software package GIMP (figure 4; [28]). The full workflow implemented here and required to introduce the pipeline into a study is summarized in figure 5.

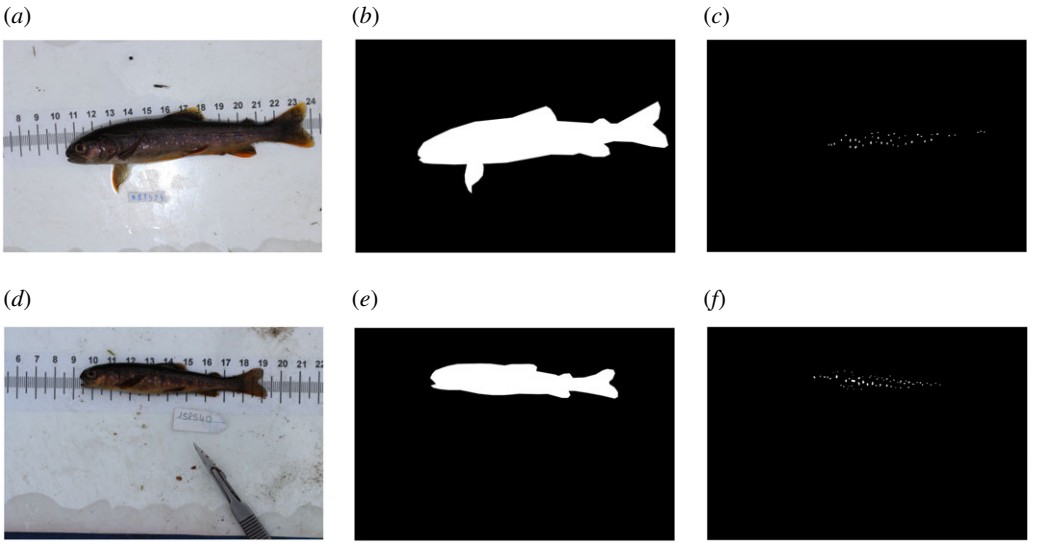

**Figure 4.** Two examples of human-annotated fish masks and spots used in the training set. Annotations were done in the photo editing software package GIMP [28]. (*a,d*) Base image, (*b,e*) mask of fish location, (*c,f*) mask of spot location.

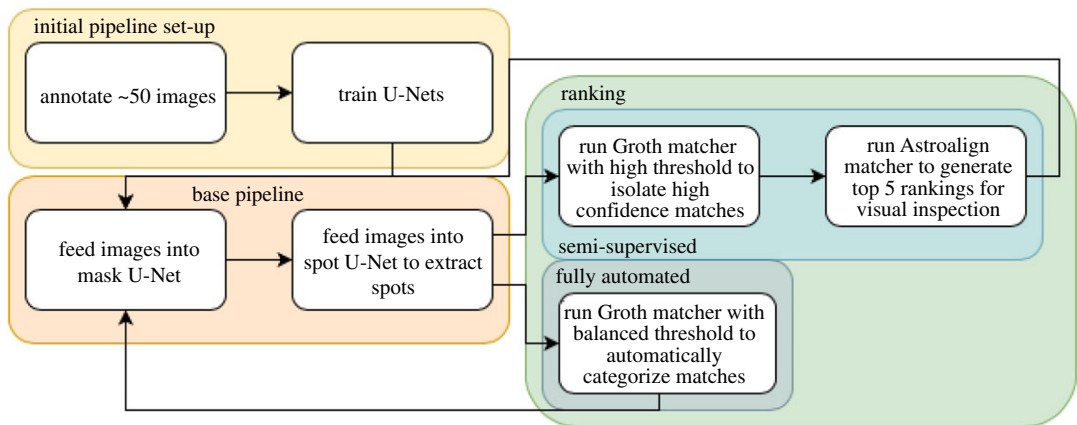

**Figure 5.** Workflow of the photo re-identification pipeline. The yellow section shows the initial pipeline set-up where the U-Nets are trained to the focal study. The orange section shows the base pipeline that is required to extract the spot constellations for matching. The green section shows the two options for spot matching, using either a semi-supervised approach that combines the Groth matcher and Astroalign algorithms (above), or a fully automated approach using the Groth matcher alone (below).

## 2.2. Spot detection

### 2.2.1. Automatic extraction of spot locations

We present and contrast two approaches to spot detection: a traditional image processing baseline method and a state-of-the-art deep-learning approach. In both cases, extraction of spots is a two-step process. We first identify the pixels belonging to the fish, which is an easier task than spot extraction and improves robustness. We then process this region of the image using a separate algorithm to detect spots. All 52 hand-annotated images were used to determine optimal parameters for the non-learning baseline. For deep-learning models, we split the dataset into training and test sets with 80% dedicated to training, resulting in 41 training images and 11 test images.

### 2.2.2. Image-processing baseline for spot extraction

Our baseline method applies a sequence of hand-crafted image processing operations to the image, as shown in figure 6. We convert the image to greyscale, apply a Gaussian blur filter ($\sigma_1 = 125$ px), and follow it by a thresholding operation that selects all pixels with a value of less than 80. The resulting

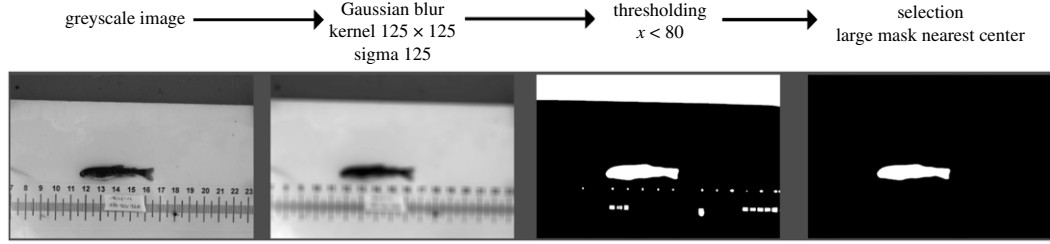

**Figure 6.** The traditional pipeline for extracting a fish mask from an image. All intensities are in the range of [0, 255].

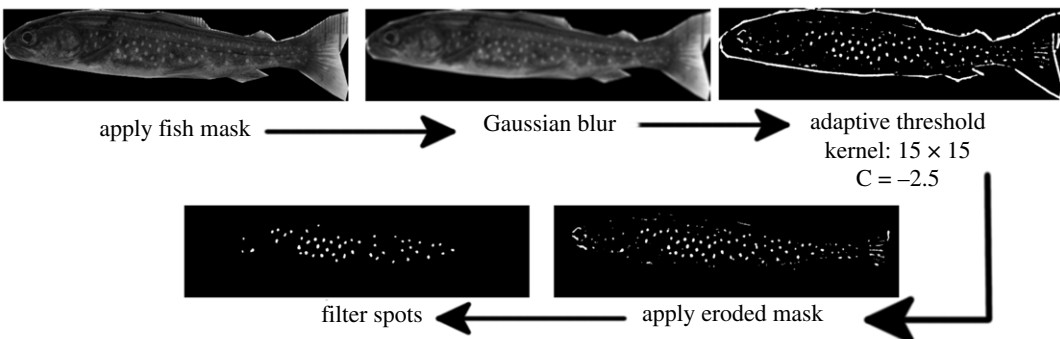

**Figure 7.** The baseline pipeline for extracting fish spots from an image.

image is processed by a connected component labelling algorithm and the largest region in the centre of the image is taken to represent the fish. This region is shrunk using a morphological erosion operator and used as a mask to concentrate the spot detection part of the baseline algorithm on the part of the image which contains a fish.

In order to detect the spots, our baseline first applies another Gaussian blur to remove high-frequency features that can interfere with detection of spots. The optimal kernel size for each fish varies, since too aggressive blurring on smaller resolution fish may cause spots to be lost. We determined that the following formula works well for our data based on empirical evaluation on the training images:

$$\sigma_2 = \left( \frac{(\text{image length} - 1882)}{440} \right) \times 2 + 19. \tag{2.1}$$

The standard adaptive mean thresholding algorithm from OpenCV is then used to identify the spots [29].[1] To remove strong responses along the edge of the fish, we then apply the mask from the previous step. Finally, we filter the remaining regions by keeping only regions that are roughly circular and medium-sized, which results in the majority of spots being extracted without a significant amount of noise (this pipeline is summarized in figure 7). This type of feature engineering approach requires domain knowledge and has many free parameters described above, such as sizes of Gaussian kernels and morphological operators, and threshold values. We have picked these parameters based on our annotated training set, but this is an involved process and it can affect the algorithm's ability to generalize to other domains or population. Still, it can be useful to contrast its performance to that of a learning-based approach described next.

### 2.2.3. Deep-learning algorithm for spot extraction

The deep-learning model is based on U-Net architecture (for example, see fig. 1 in [30]) and was chosen for its success in biomedical and binary image segmentation [31], which often involves segmenting blob-like structures from images. The architecture of the U-Net used in our work consists of four hidden layers for the encoder, with 64 filters per layer and a 50% dropout per layer in the encoder. The input was a 512 × 512 RGB image, and the output was a 512 × 512 greyscale image. The input images were scaled using black padding on the edges to fit the input size, to maintain the aspect ratio of the image and

---

[1]In our experiments, adaptive thresholding significantly outperformed standard thresholding because lighting conditions can vary significantly between photos due to reflections on fish scales.

avoid distortions. The network was trained for 150 epochs, using 50 images per epoch. Standard data augmentation techniques were applied to the input data in the form of random rotations, shears, zooms, flips and brightness adjustments.

The same U-Net architecture was trained separately on the two sub-tasks. First a network was trained to extract the fish mask using the original images from the annotated training set. In order to train the second network to extract spots, simple pre-processing was needed since down-sampling to $512 \times 512$ pixels can lead to smaller spots being missed. In this case, the original image was first cropped to the extent of the mask extracted by the first network, and rescaled so that the largest dimension is scaled down to 512 pixels, preserving the aspect ratio. Empty space was padded black. This resized and padded image was then passed to the network for training.

### 2.2.4. Evaluation of experimental results for spot detection

Fish mask extraction and spot detection were evaluated on the test set of 11 annotated images. In each case, we evaluated per-pixel precision and recall, and calculated the F1-score [32] to evaluate how well the predicted masks match the annotations. We show later that the level of accuracy achieved was sufficient for successful re-identification of fish.

## 2.3. Matching spot patterns

Once the spots are extracted into $x$-, $y$- and size features by finding the contours of the generated spot image using the OpenCV library [29], several algorithms were tested and the two best-performing algorithms are summarized below. Both algorithms are based on forming and characterizing triangles from triples of points in two-dimensional space.

### 2.3.1. Custom Groth matcher

The Groth matcher algorithm has multiple steps: spot selection, triangle generation, triangle matching and assigning matched points. It assumes a certain tolerance range for locations, $\epsilon$ and any point within three $\epsilon$ from another point in the same list is removed from the list. Next, all possible triangles from the points in each list are created. For each triangle, it calculates: the logarithm of the perimeter, orientation of traversing the shortest to longest side, ratio of longest to shortest side, the tolerance in the ratio, the cosine of the angle between the longest and shortest side and the tolerance in the cosine. All triangles with a side ratio greater than 10 were removed, as the ratio tolerance significantly increases when the ratio increases. Due to the same reasoning, all matches with a cosine value of greater than 0.99 were also rejected. A match is made between two triangles when the square of the ratio difference between the triangles is less than the sum of the squares of the ratio tolerance and the square of the cosine difference between the triangles is less than the sum of the tolerances in the cosine values. When multiple matching triangles are present, the one with the lowest difference in the two features is selected.

To reduce the number of false matches, the logarithm of the magnification factor is computed from the difference in logarithms of the perimeter between each match. Any match landing outside a standard deviation from the mean is rejected. Additionally, matched triangle orientations are compared with determine whether their mapping agrees with the majority of other triangle matches. The minority mapping is rejected.

A voting system is used to determine which points are the most probable matches. Each matched triangle casts a vote for each pair of its matched vertices. The paired vertex list is then sorted in descending number of votes. Each pairing is assigned as a correctly matched pair successively until: the vote count drops by a factor of 2 from the first pair, an attempt is made to assign a point that has already been assigned or the vote count drops to 0. The whole algorithm is then repeated, but only with the points from the matched pairs to avoid spurious assignments. If the second round of the algorithm returns the same number of matches, the original matches are assumed to be correct. If fewer matches are returned than the first round, the spots are considered impossible to match.

We based our work on Pascual's [33] implementation, but modified it to match the implementation described by Arzoumanian *et al.* [17]. We kept the k-d tree optimization introduced by Pascual [33], but switched the k-d tree from a Python-based implementation to a C-based implementation for speed. As suggested by Arzoumanian *et al.* [17], we added the angle between the first vertex (the vertex between the longest and shortest side), the centroid of the triangle and the horizontal as an additional feature to

help improve matching. However, we discarded the restrictions of the cosine angle used by Arzoumanian *et al.* [17] and the maximum triangle-side length ratio from the original Groth algorithm. Our rationale was that Arctic charr spot distributions are often organized into very linear patterns, so these restrictions excessively restricted the number of triangles selected on smaller specimens. Instead, to limit the number of triangles created, we only considered the nearest 25 points when creating triangles, which also increased the speed. The restriction on the maximum angle difference implemented by Arzoumanian *et al.* [17] was kept at 10°.

The biggest restriction on performance of the Groth algorithm is the way matching is performed between triangles. Both the method used by the original paper [26], and the quadrature distance used by Arzoumanian *et al.* [17] requires $O(nm)$ comparisons, where $n$ and $m$ are the lengths of each spot list respectively, to determine the match list. Instead, we used a k-d tree for each list with the features space of the side ratio, cosine of the angle, and the angle to the centroid to insert triangles into. The two k-d trees were then compared to find triangles which are mutually closest to each other in the feature space and then taken to be the initial matches. This results in a faster operation, as it only requires the nearest neighbour to a point in a k-d tree to be found, which results in a total of $O(\log(n)\log(m))$ comparisons. As the algorithm does not need to deal with reflections or inversions of fish spots, the matches were then filtered so that only same-handed triangles were selected. These were filtered to have the difference in the logarithm of the perimeter be within 1.5 standard deviations of the group mean. At this stage, the algorithm votes and selects points as normal and repeats the process with the selected points. The scoring was performed in the same way as the standard Groth matcher [26].

## 2.3.2. Modified Astroalign

The Astroalign (AA) library by Beroiz *et al.* [27] takes inspiration from the Groth algorithm, but also implements a random sample consensus approach (RANSAC) to selecting the transformations over Groth-style voting. This has the benefit of more directly taking into account that a number of spots will always be outliers in every comparison. RANSAC is commonly used in computer vision to find a homography between two sets of points. It increases the robustness of a model fit by randomly selecting points on which to fit a model, before assessing the model on the remaining points for how many good (inlier) and bad (outlier) matches there are [34]. This process is repeated until the proportion of inliers exceeds a threshold (as described later in this section), or it hits a set number of iterations. Fish can develop new spots over time and old spots can change, so even between images of the same individual there will always be outlier spots in the data.

The AA algorithm builds on top of triangles with two invariants:

$$\frac{L_2}{L_1} \tag{2.2}$$

and

$$\frac{L_1}{L_0}, \tag{2.3}$$

where $L_2 \geq L_1 \geq L_0$ are the lengths of the sides of the triangle.

To these two values, we added the angle between the vertex at the intersection of lines $L_0$ and $L_1$ and the centroid of the triangle. This increased the discriminatory power of the invariants and restricted excessive deformation. In our work, the masks of the two fish were roughly aligned in pre-processing so the spots should not have to move far to find a match. Lastly the $x$- and $y$-coordinates of each vertex were added (down by a factor of 5 to match typical magnitudes of other features), which encourages matching of triangles in the same area of each fish. This also had the effect of reducing the likelihood of considering spot alignments that did not align the masks properly.

A local approach was used for triangle construction, with the closest 15 points being used to generate triangles, rather than all the possible points. The biggest deviation from the Groth matching algorithm was how the matches were evaluated. In the RANSAC step it randomly selected a matched triangle. Then it used the three pairs of vertices to calculate a transformation. This transformation was assessed by calculating the Sampson distance [35] of all point pairs in other matched triangles and selecting the highest error of the three vertices for each matched triangle. Each matched triangle is then determined to either be an inlier or outlier by applying a threshold to the error of each matched

triangle. If the number of inliers exceeded 85%, then all inlier matched triangles were used to generate a refined transformation using all available data.

We applied a further constraint by limiting allowed rotation, scaling, and translation between the two sets of points. We measured this through a transformation disturbance defined as

$$\frac{\text{rotation}}{60} + \frac{\text{abs(scale} - 1)}{2} + \frac{\text{Manhattan displacement}}{2}. \tag{2.4}$$

This again restricted the solutions to the ones that had a high overlap of fish masks as large transformations were discouraged because the fish masks were roughly aligned in the pre-processing step.

If the transformation did not terminate RANSAC early, it was stored if it had the smallest transformation disturbance compared with previous transformations which reached the 85% inlier threshold but not the maximum transformation disturbance threshold. This was to ensure that a match could be assigned a score even if it did not meet the strict criteria, such as when comparing fish with significant growth between images.

The score was then calculated as the percentage of inliers in the target points multiplied by the percentage of inliers in the source points. Once the score and list of inliers was retrieved, the mask F-score was calculated by applying the resulting transformation to the fish mask, and overlaying over the target mask. All solutions with F-scores below 0.75 were rejected regardless of the actual score.

### 2.3.3. Evaluation of experimental results for spot matching

The matching performance of the two algorithms was evaluated using two accuracy metrics: open-set and closed-set accuracy. Open-set accuracy is the percentage of fish either correctly matched to an existing specimen, or correctly identified as not having a match. Closed-set accuracy only considers cases where there exists a match for a given individual in the database already. Open-set accuracy is more relevant to practitioners in the field because it is often the case that there is no match in the database and it would be useful to detect this automatically. However, most matching algorithms are designed to find a match that exists, and focus on ranking. By reporting closed-set accuracy, we evaluated how well these algorithms translate to the spot-matching problem.

When reporting open-set accuracy, we reported the number of true positives, false positives, true negatives and false negatives for both algorithms. A true positive is defined as a correct top-1 match. A false positive is a fish which is incorrectly matched to a different individual. A true negative is a correctly classified lack of match, meaning that the algorithm correctly identified that the individual had not been seen before. A false negative occurs when an individual with a match in the database was incorrectly identified as an individual which does not exist in the database, giving no results. When reporting closed-set accuracy, we reported rank-1, rank-2 and rank-5 accuracy, which considered a match to be correct if it was ranked within the top 1, 2 and 5 matches, respectively. While rank-1 accuracy is the best case, rank-5 accuracy is still useful for flagging up possible matches for an expert to confirm manually.

The inputs to each test consisted of the spots and masks generated by the U-Nets in previous steps. This was to make sure that the algorithm performs well with the data generated by the previous stages in the pipeline, rather than the 'ideal' annotated case. This provided a more realistic approximation of real-world error for the entire matching pipeline.

To improve the performance of the algorithms, we performed an additional pre-processing step to correctly align the fish before being passed to the algorithms. This was done by rotating both the spots and fish mask such that the mask's primary components (determined by primary component analysis; PCA; [36]), were aligned with the $x$- and $y$-axes, respectively. Then, the coordinates of the spots were scaled to the length of the fish so that all values fit between 0 and 1. This ensured that the initial position of the spots was fairly close to where they should be present on the other image.

To compute the accuracy metrics in a more realistic scenario the fish photos from each cave tested were split into collection dates. Then, each collection was processed one at a time and added to the database, so that each individual in a collection was only compared with images that were taken before it was photographed. This mimics the real-world scenario, where previous captures of a fish will always be the same or smaller.

## 2.4. Survival estimation as a case study comparison

One common use of CMR studies is to estimate survival rates. Therefore, to evaluate the contribution of our photographic identification pipeline to survival rate estimation in the Arctic charr study system, we

(a)   (b)   (c)

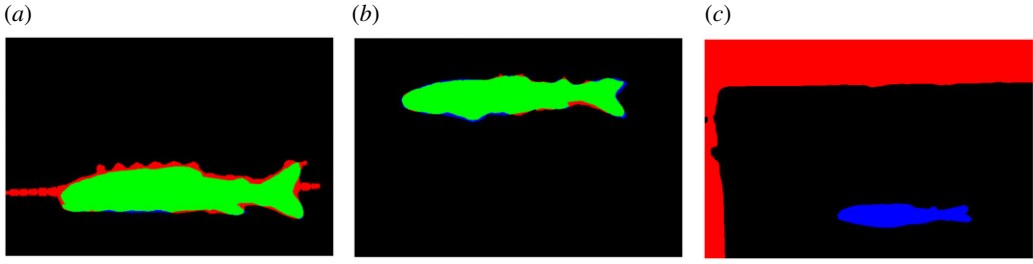

**Figure 8.** Examples of segmentation results from baseline method testing: (a) a ruler captured in the segmentation, decreasing precision; (b) correct segmentation of a fish; and (c) image not centred correctly on the fish, resulting in a segment of the background being incorrectly identified as the subject of the image. In all examples, blue represents false negatives, green represents true positives and red represents false positives.

fitted a simple Cormack–Jolly–Seber (CJS) CMR model [37] to two versions of the capture histories from Caves A and B from August 2012 to 2019 (excluding the first capture occasion), and a third including the first capture occasion. First, we used capture histories for these fish based on the PIT tag data alone ($n = 291$; $n$ refers to number of individuals), which we originally suspected would contain fragmented histories due to tag loss. Second, we used the most complete version of the capture histories possible ($n = 265$), where the PIT tag data were corrected for re-identified individuals from our photographic tool.

The survival models took the form

$$a_{i,t} \sim B(a_{a,t-1} \cdot \phi) \tag{2.5a}$$

and

$$y_{i,t} \sim B(a_{a,t} \cdot p), \tag{2.5b}$$

where $a_{i,t}$ are the (partially observed) state variables describing whether individual $i$ is alive at time $t$ and $y_{i,t}$ are capture histories ($y_{i,t} = 1$ if individual $i$ was captured at time $t$ and zero otherwise). $\phi$ is the per-interval survival rate and $p$ is the capture probability. $B(a)$ represents a sample from a Bernoulli distribution (binomial distribution with one trial), with a probability of success of $x$. The likelihood of the model defined by equations (2.5) is evaluated over all capture occasions subsequent to each individual's first capture. We sampled the posterior distribution of a Bayesian model defined by equations (2.5), with uniform priors between zero and one for $\phi$ and $p$, using JAGS [38].

By including the first capture occasion (June 2012) re-identified individuals were classified into those that were not related to tag loss (category 1) and those that were (category 2). Category 1 includes individuals that were never tagged in the first capture occasion but were photographed. There were also a few individuals that were re-identified as discrepancies in data entry. The individuals in category 2 gave us an estimate of the extent of tag loss within the system, and of the practical consequences of correcting for tag loss. This categorization resulted in 17 individuals in category 1, 16 in category 2 and 261 not categorized for the new capture histories across both caves (i.e. individuals that had complete capture histories in the uncorrected data as far as we can tell). We re-ran the model using this dataset with corrected capture histories including the first capture occasion ($n = 294$).

# 3. Results

## 3.1. Experimental results for spot detection

The U-Net performed substantially better than the baseline method in fish mask extraction, achieving an F1-score of 0.972 compared with an F1-score of 0.902, respectively. The baseline struggled when the fish was positioned off-centre or when the fish was surrounded by significant debris, as seen in figure 8. While the U-Net already achieved good results, we note that masking can be slightly improved by simple post-processing. For example, by flood filling the internals of the prediction to avoid holes and using a morphological closing operation to join up disjoint regions (figure 9). The result could also be improved by training on a wider variety of individuals. Overall, the U-Net is a promising approach to extract fish masks, as it can be trained to a high level of accuracy on a small number of individuals using image augmentations. It also indicates that this network is well suited to detecting the intended

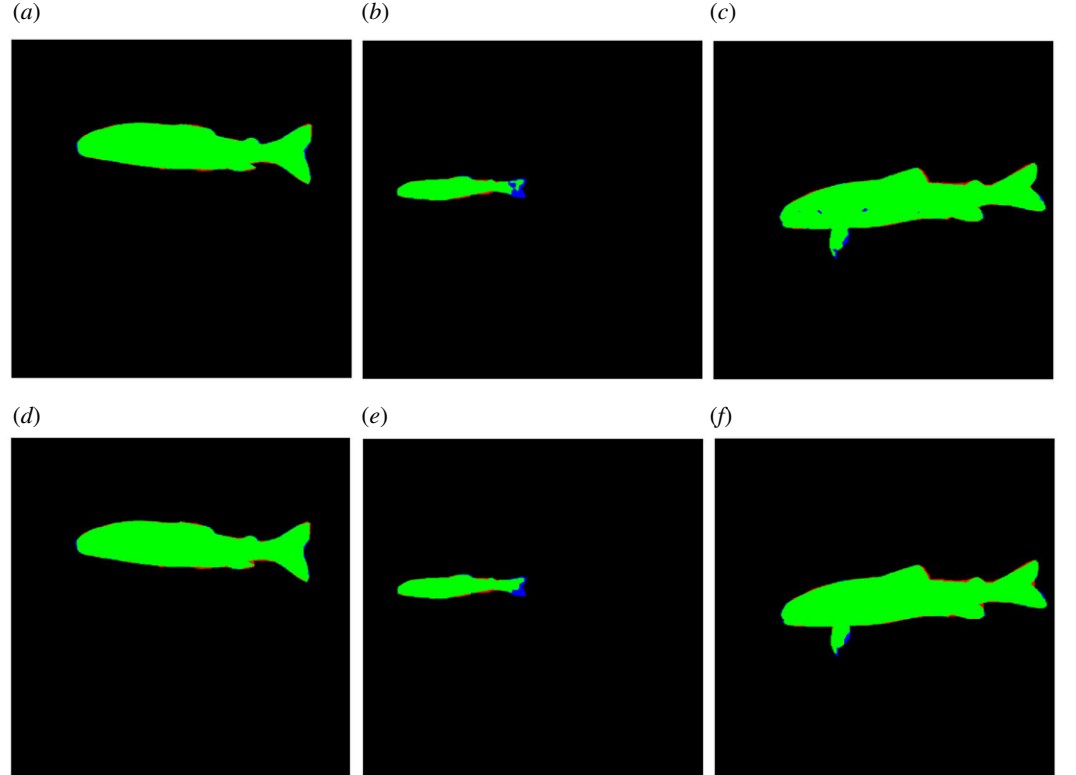

**Figure 9.** Examples of segmentation results from the U-Net testing set: (*a*) an example of a high-quality segmentation; (*b*) an example of poor segmentation of the tail; (*c*) an example of holes in the prediction; (*d*) image a after post-processing; (*e*) image b after post-processing; (*f*) image c after post-processing. The top row images (*a–c*) are before post-processing, and the bottom row images (*d–f*) underwent post-processing operations of flood filling shapes, erosion and dilation and isolation of the biggest central mask. In all examples, blue represents false negatives, green represents true positives and red represents false positives.

subject of the image, even in situations where the fish is off centre or on top of high contrast markings, such as a ruler.

On the task of spot extraction, our baseline achieved an F1-score of 0.418, performing relatively well on large fish with clear spots, but poorly when there were bright spots present, such as around the operculum. The U-Net architecture achieved an F1-score of 0.566, dealing much better with variation in lighting and colour of the spots (figure 10). The baseline algorithm suffered from a large number of false negatives, and the false positives were spurious in occurrence, especially near the head and the tail. By contrast, the U-Net architecture exhibited far fewer false negatives (thus successfully identifying the majority of spots labelled by a human), while its false positives often corresponded to spots that were missed from manual labelling. Here we note that, while these F1-scores appear to be low, the score is evaluated at the level of pixels where we would not necessarily expect high correspondence due to the inaccuracy of manual spot annotations which cannot be pixel-perfect. This affects spot extraction more than the fish mask extraction task due to the larger number and smaller size of the regions.

## 3.2. Output of the pipeline

Both algorithms performed well and managed to discover multiple mismatched tags in the dataset. We found 20 mismatched fish in Cave A and 22 in Cave B. Of these mismatches, 10 and 12 were PIT tags that fish had lost in Cave A and B, respectively, which relates to 6.5% PIT-tag loss in Cave A and 7.5% PIT-tag loss in Cave B. The remaining mismatches were individuals re-identified from the first capture occasion who had never received a PIT-tag, or data entry mistakes; 10 individuals in Cave A (6.5%) and 10 in Cave B (6.3%). The custom Groth matcher performed better and more consistently than the AA matcher, especially when it comes to false positive rates (table 1). Additionally, the threshold of 6.5 for score matches (where any score above the threshold was accepted as a correct match and anything

| fish specimen | baseline spot segmentation | U-Net spot segmentation |
|---|---|---|

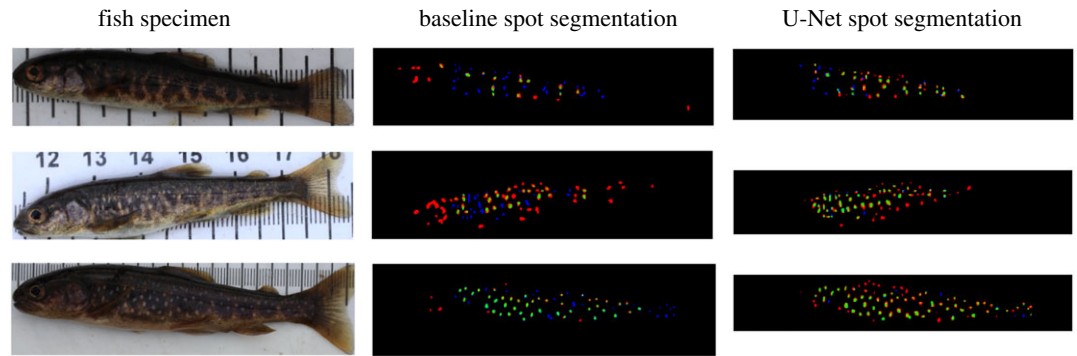

**Figure 10.** Comparison of baseline and U-Net spot segmentation on various sizes of fish. A selection of results are shown, with correctly classified pixels shown in green, and errors shown in blue—false negative, or red—false positive.

**Table 1.** Results of matching from the modified Groth matcher (*a* and *b*) and modified Astroalign algorithms (AA; *c* and *d*) showing the open-set accuracy of each. These matrices show the number of true positives (a correct top-1 match), true negatives (correctly identified as having no matches), false positives (incorrectly matched to a different individual), and false negatives (incorrectly identified as having no match) for each matching algorithm in Caves A and B. The totals for the true classification of images are shown in italics, alongside the total number of false classifications.

| | positive | negative | total |
|---|---|---|---|
| (*a*) Cave A modified Groth matcher | | | |
| true | 97 | 80 | *177* |
| false | 4 | 54 | 58 |
| (*b*) Cave B modified Groth matcher | | | |
| true | 113 | 140 | *253* |
| false | 3 | 68 | 71 |
| (*c*) Cave A modified AA | | | |
| true | 104 | 70 | *174* |
| false | 16 | 45 | 61 |
| (*d*) Cave B modified AA | | | |
| true | 80 | 131 | *211* |
| false | 18 | 95 | 113 |

below the threshold was rejected) determined to work optimally in Cave A was also the optimal threshold in Cave B.

The custom Groth matcher had a very sharp rise in score when a match was found, resulting in a higher number of true positives in both caves (table 1*a,b*) with no human input. In this mode of operation, it managed to achieve 75.3% open-set accuracy in Cave A and 78.1% in Cave B. However, in a semi-automated setting with a human in the loop, the AA algorithm performed significantly better. Without a threshold and having a human select from the top few ranked matches, 91.5% of the correct matches from the AA algorithm were in the top 5 (ignoring images with no match), and of these 84.3% were the top match for Cave A (10; figure 11). The grouping of the matches in higher ranks was especially evident in Cave A, but was also seen in Cave B, where a significantly higher proportion of matches were closer to the top, with fewer matches being completely rejected and not being assigned a score.

The size of the fish being matched had an effect on how well the algorithms performed. A greater size difference between two images of the same fish decreased the overall accuracy of matching for both algorithms (figure 12). However, whereas using the AA algorithm resulted in a decrease in the scores for invalid and correct matches as the size difference increased, using the custom Groth algorithm only resulted in correct match scores dropping while the invalid scores did not. The drop in the

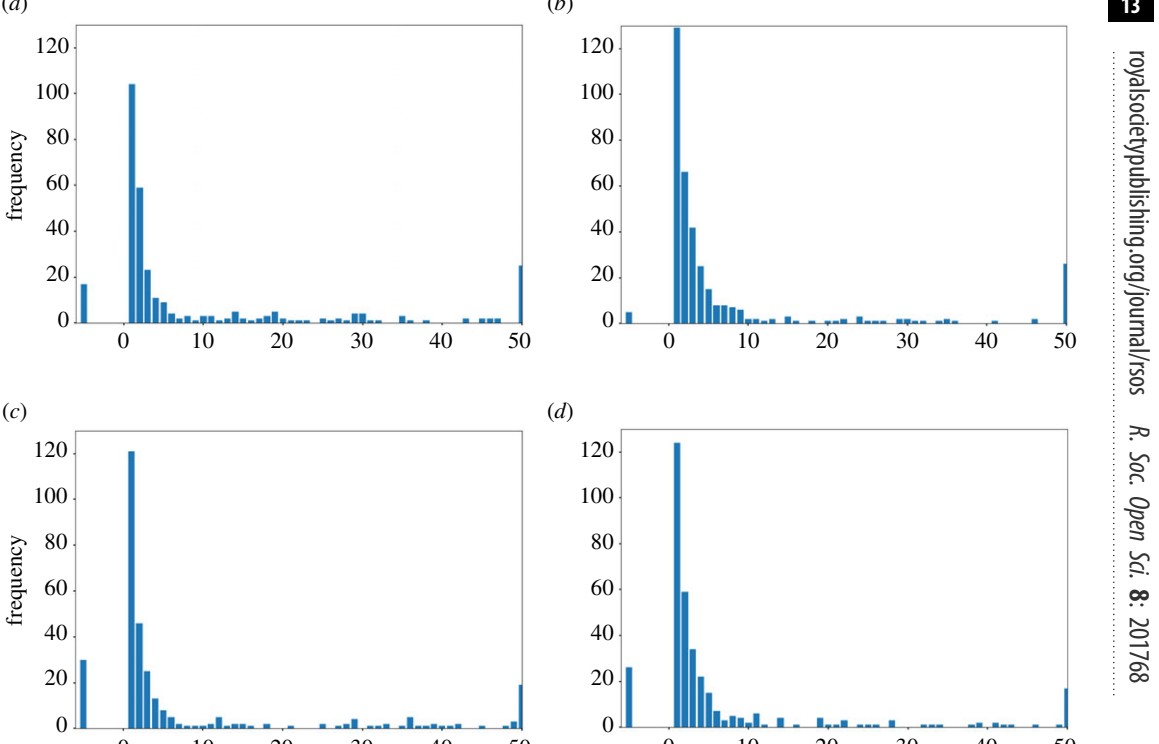

**Figure 11.** Frequency of a matching fish occurring at a given rank. All matches that were ranked at rank 50 or below have been bucketed into the bar shown at rank 50. All matches that failed to be ranked are summarized in the bucket shown at −5. (*a*) Cave A modified Groth matcher, (*b*) Cave A modified AA, (*c*) Cave B modified Groth matcher, (*d*) Cave B modified AA.

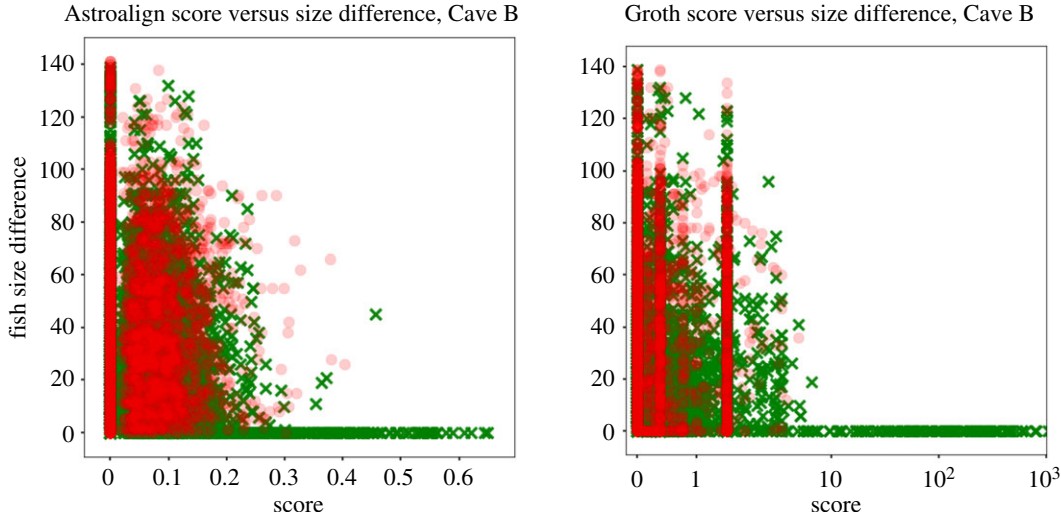

**Figure 12.** The effect of fish size difference on the score from the two matching algorithms using fish from Cave B, with the custom Astroalign and the custom Groth matcher results shown on the left and right, respectively.

custom Groth matcher performance is due to the lower number of spots present on smaller fish. This in turn restricts the maximum number of triangles that can be matched, making it harder to distinguish between random matches and true matches. Comparatively, the AA algorithm performs better on smaller fish and pairs of fish with larger size disparities for two reasons: it performs a secondary step for validation of the match using the fish mask (which is difficult to replicate in the custom Groth matcher), and it is based on the more computationally intensive but more robust RANSAC algorithm.

The accuracy differences resulting from trying to match different-sized fish to the database highlighted the drop in accuracy of both algorithms in fish smaller than 110 mm, especially for the

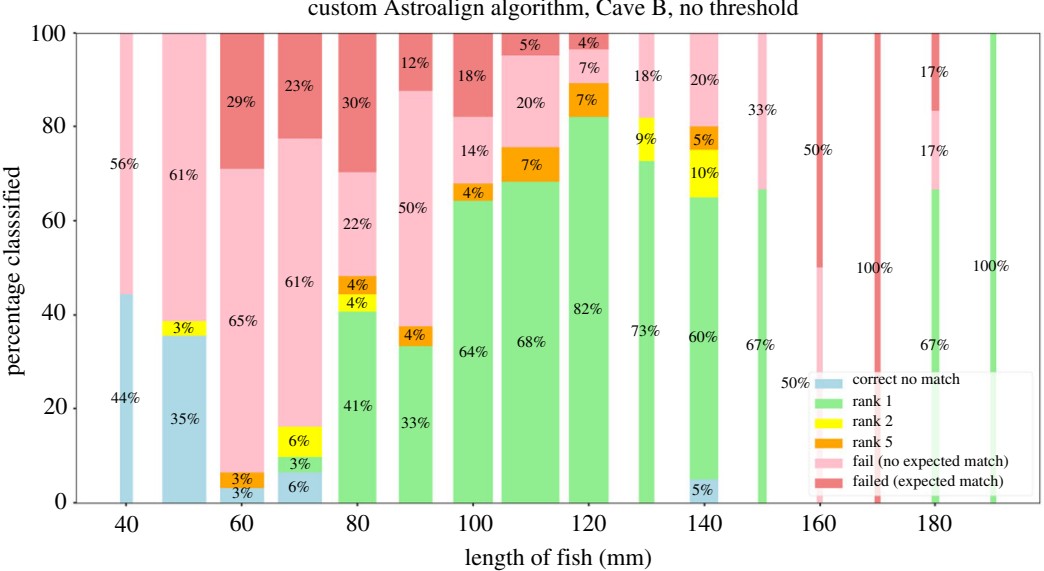

**Figure 13.** Performance of the custom Astroalign algorithm based on size of fish being matched. The width of the bar represents the size of the bin relative to the rest of the bins.

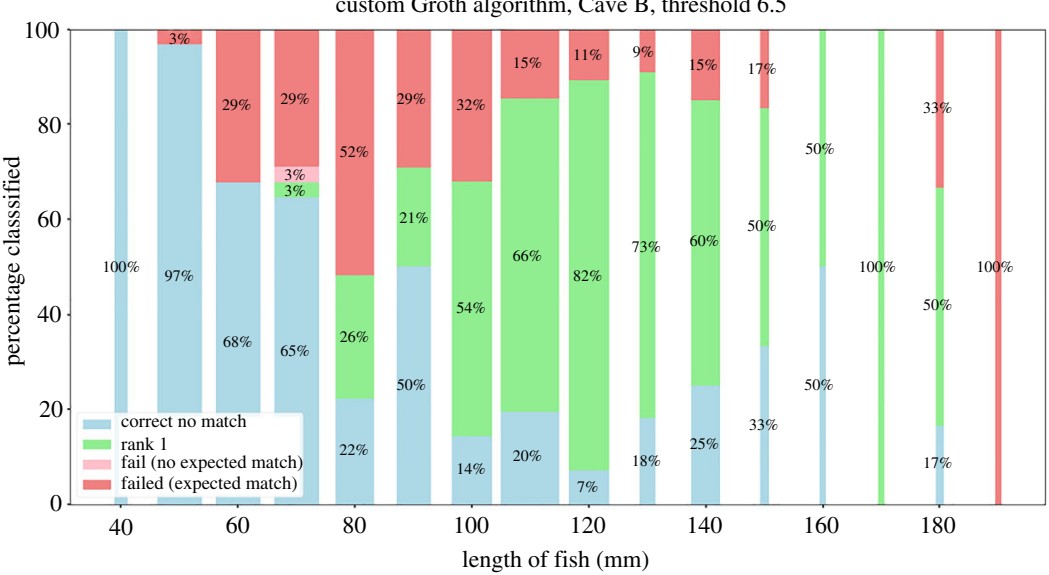

**Figure 14.** Performance of the custom Groth algorithm based on size of fish being matched. The width of the bar represents the size of the bin relative to the rest of the bins. A threshold such that all scores below 6.5 were rejected was applied to automatically categorize fish with no match in the dataset.

custom Groth matcher (figures 13 and 14). The results show that while the AA matcher performs worse in open-set top-1 accuracy, the majority of the invalid matches are ones where the fish had no actual match in the database. The AA matcher performs much better in the closed-set scenario, especially for smaller fish, meaning that the relative ranking for each fish remains fairly stable as spot numbers and fish size decreases. We can conclude that the AA matcher is better at ranking fish, especially in the presence of large size disparity, but is considerably less accurate at detecting if a fish does not have a match.

In order to combine the relative strengths of the two algorithms, we explored an ensemble of the two algorithms by applying a threshold on the results of both algorithms. If full automation is still desired, the scores can be combined using a simple OR operation after thresholding the score outputs of each algorithm separately. For both caves, using this approach increased the rank-1 open-set accuracy to 81%.

**Table 2.** Survival ($\phi$) and capture rate ($p$) estimates from capture–mark–recapture models applied to: (*a*) data from August 2012–2019 with suspected tag loss ($n = 291$); (*b*) data from August 2012 to 2019 where individual identification is supplemented by photo-identification ($n = 265$); and (*c*) data from June 2012 to 2019 where individual identification is supplemented by photo-identification ($n = 294$). Parameter estimates (posterior means) are given with limits of the region of 95% highest posterior density.

| parameter | estimate |
| --- | --- |
| (*a*) original capture histories: August 2012–2019 | |
| $\phi$ | 0.74 (0.71–0.77) |
| $p$ | 0.53 (0.48–0.57) |
| (*b*) corrected capture histories: August 2012–2019 | |
| $\phi$ | 0.78 (0.75 − 0.81) |
| $p$ | 0.53 (0.49 − 0.58) |
| (*c*) corrected capture histories including June 2012 | |
| $\phi$ | 0.77 (0.74–0.80) |
| $p$ | 0.49 (0.44–0.53) |

## 3.3. Consequences for estimation of survival rate

There was an approximately 4% increase in the mean estimate of survival rate ($\phi$; table 2) when using capture histories corrected for tag loss from the photo re-identification pipeline (data August 2012–2019). Including individuals identified from photographs in June 2012 gave a slightly lower survival rate ($\phi = 77\%$) than with corrected capture histories from August 2012 ($\phi = 78\%$); this is still an approximately 3% increase on the uncorrected estimate ($\phi = 74\%$). The mean capture probability did not change by correcting the capture histories from August 2012–2019 ($p = 53\%$; table 2*a*,*b*), however, this value decreased when individuals from June 2012 were included ($p = 49\%$).

# 4. Discussion

The tool developed here to re-identify individuals based on spot constellations in photographs performed well in our study system, compared with a baseline approach, and should be transferable to other systems wishing to match images based on spots. A key component of the pipeline is tuning it to the focal system by first training the algorithms on a small set of expert annotated images. This, alongside the ranking which allows for simple human comparison and confirmation of matches post-processing, gives high confidence in re-identified individuals. This work highlights the significant improvements to time and accuracy of the re-identification process that can be obtained by combining a greatly reduced requirement for human supervision with deep learning over using either method separately. We also show that the pipeline maintains a high degree of accuracy if used in a fully automated manner post-training, i.e. without post-processing human match confirmation, because of the precision of top-1 ranked matches using the Groth matcher algorithm on its own. Furthermore, we demonstrated that accurate re-identification had an impact on an important aspect of biological studies—the estimation of survival rates—in our study system, with an increase of approximately 4% found for fish survival. Overall, this publicly available tool and innovative way of thinking will be useful for re-identification of individuals from spots in images for a wide range of study systems.

## 4.1. Performance and transferability

Compared with baseline approaches, our tool performed better in both the fish masking (baseline F-score = 0.902; U-Net F-score = 0.972) and spot extraction steps (baseline F-score = 0.418; U-Net F-score = 0.566). Although the performance of the baseline approach was passable, that type of feature engineering approach requires substantial domain knowledge and it is time-consuming to optimize the parameters to a specific study system. The difference in performance between the baseline and our approach was most obvious when the image quality was sub-optimal, which is inevitable in most field studies (e.g. there will be some images with debris in them or an individual positioned off-

centre). These imperfect images significantly affected the performance of the baseline approach (figure 8), but not the U-Net approach used in our pipeline, which was able to recognize the intended subject of the image even with disturbances within the photograph. Furthermore, despite performing relatively well on large fish with clear spots, when bright areas were present on the fish (e.g. around the operculum), the baseline performed poorly, suffering from many more false negatives (figure 10). These issues in the baseline algorithm's ability to capture the fish mask and correctly identify spots could result in a substantial loss of information in a study system. The accuracy of the spots being extracted by the U-net could potentially be improved by using image patches, resulting in a higher resolution of the fish being used in segmentation. However, the results achieved using the method outlined here were shown to be sufficient for matching, while using less processing time. This is because in order to have $512 \times 512$ patches per image, the U-Net would need to be run around 60–77 times per image instead of once. Therefore, depending on the image sizes (most here were $5184 \times 3456$) processing time is sped up by roughly that factor by not using patches. This is a good example of optimizing the output of our pipeline while minimizing the resources used. Therefore, the baseline is both slower to set up, more prone to data loss and less applicable for generalizing to other study systems relative to our learning-based approach.

When applying our tool to different systems, a simple point to remember for spot extraction would be that using a human-annotated training set of around 50 individuals that represent the variation within the study system will greatly improve the speed and accuracy. While the annotation process could be considered slow and requires some expert knowledge, we note that this is a very small training set compared with typical computer vision datasets which often require tens of thousands of annotated images [16]. Although we expect that a smaller number of images could be used depending on the variation within the study system, annotating 50 images takes relatively little time.

In general, both of the customized spot matching algorithms performed well, with different advantages and disadvantages. The custom Groth matcher would be a better choice for a fully automated matching because the sharp rise in score when a match is found means that each top match gets either accepted or rejected based on a preset threshold with no human input. This meant that the Groth matcher had a higher number of true positives in both caves (table 1) and greater open-set accuracy compared with AA algorithm (the Groth matcher achieved 75.3% in Cave A and 78.1% in Cave B; the AA algorithm achieved 74% in Cave A and 63.9% in Cave B) in the automated setting. However, in a semi-automated setting with a human in the loop, the AA algorithm performs better. This is because the AA algorithm, while not having a fairly well-defined threshold to reject matches, ranks matches better for each fish. This means that when no threshold was used and a human was asked to select from the top few ranked matches in Cave A, 91.5% of the correct matches were in the top 5 (ignoring images with no match), 84.3% of which were the top-1 match (figure 11). This demonstrates that while top-1 accuracy is the best case, top-5 accuracy is useful for flagging up possible matches for an expert to confirm manually in this semi-automated scenario. The different pros and cons of the two matching algorithms are a great strength of our pipeline, giving the user flexibility between semi- or fully automated implementation and still improving identification of individuals.

In a semi-automated scenario, an ensemble of the algorithms can be used to reduce the amount of human effort spent on checking matches. By using the custom Groth matcher's very good false positive rate, it can be used to automatically categorize all strong matches without human intervention. The remaining samples can then be passed on to the AA-based matcher and human verification. This would leave a significantly smaller number of fish that would have to be verified by hand, while significantly improving overall accuracy, as it combines the Groth matcher's excellent open-set matching capability with the AA's strong closed-set ranking capabilities. We tested various other algorithms normally used for co-registration of point clouds, for example, the iterative closest point (ICP; [39]), but these gave surprisingly poor results and so we do not present that work here. We believe this is due to fish having large discrepancies in the number of spots present, which lead to significant warping of the fish during matching, or getting stuck in local minima.

## 4.2. Application to the case study

The photographic re-identification tool allowed us to correct and identify a substantial number of capture histories in the Arctic charr study system (13% in Cave A and 13.8% in Cave B; including June 2012). There were two types of re-identification present in this case study, both of which are relevant to individual-based studies, and were comparable in magnitude. The first type was discovery of fish that

were never PIT-tagged from the first capture occasion, plus a few data entry mistakes (6.5% in Cave A and 6.3% in Cave B). This demonstrates the ability of the pipeline to use photographs of individuals that are not tagged and identify matches through time. Furthermore, as the pipeline re-identified a few individuals that were data entry mistakes, it could be time-efficient to allow a pipeline like this to quality check data. In general, in cases where it is not possible to tag individuals regularly, such as endangered or highly cryptic species, this type of photo-ID system could enable an individual-based study to occur where it might otherwise be impossible. It is important that we continue to refine methods of individual identification to be less invasive, which aligns our method with the 3Rs framework (replacement, reduction and refinement; [40]) of ethics in animal welfare. Although the quality of photographs and phenotypic measures of interest will also have an impact on methods, removing the need to tag individuals post-capture would be beneficial in many cases (e.g. [41]). Therefore, as a tool for improving confidence in individual identification data without dramatically increasing the human-driven processing time, this method of photo-ID could prove useful.

The second type of re-identified individuals were apparent cases of tag loss (6.5% in Cave A and 7.5% in Cave B), which is a known issue even with modern tagging methods [6,8]. Being able to correctly estimate tag loss in a system is a great advantage when running CJS models as such estimates can be used to remove bias that occurs through assuming perfect tagging [37]; for example, if a tag is lost and it is assumed that the individual has not been recaptured, that individual will be thought to have died or emigrated, leading to a decrease in the estimate of survival rate. Using corrected capture histories in the Arctic charr system increased the estimates of survival rate by approximately 4% (table 2), which could be consequential for management as population growth rate and viability are often sensitive to even small differences in survival rate (e.g. [42–44]). This is particularly the case for long-lived species where mean longevity and consequently reproductive output can change greatly for small changes in per-interval survival.

Resolving recaptures that had not previously been recognized due to tag loss might be expected to increase the estimate of the recapture rate. However, the recapture rate estimated from the original data, and the data with resolved IDs, were very similar (table 2a,b). This occurs because of the concomitant rise in the estimate of the survival rate. Consequently, there is an overall tendency for the model of data with resolved IDs to expect longer survival following last captures; this corresponds to longer intervals post last capture, and explains why the estimate of the capture rate does not increase. In general, the assessment of vital rates, such as survival, on population dynamics and the implications for management strategies is an active field of research because there is no definitive method to assess their impact across systems [45], which could be partly driven by uncertainty in the estimates of vital rates themselves. Therefore, a simple method of re-identification that improves the accuracy of survival rate such as ours will be valuable for population management.

Overall, we have demonstrated the feasibility of re-identifying individuals from photographs using these spot detection and matching algorithms. We have also shown how our re-identification method makes it possible to increase sample sizes without a great deal of extra effort in terms of data processing. This could be used to deal with different types of re-identification requirements. These results are promising and open up possibilities for re-identification from spots not only in these Arctic charr, but, in the longer term, in other species that exhibit spots. Finally, our results show that this tool can be used to support individual-based studies where tag loss is an issue, by identifying its extent and reducing its effect.

Data accessibility. Data and relevant code for this research work are stored in GitHub: https://github.com/eamittell/Arctic_charr_re_id.git and have been archived within the Zenodo repository: https://doi.org/10.5281/zenodo.4981128.

Authors' contributions. M.B.M. and K.T. conceptualized the study; C.A.L., B.K.K. and E.A.M. collected the data; I.T.D. and K.T. wrote the code for the pipeline; E.A.M. and M.B.M. analysed the biological data; I.T.D., E.A.M., M.B.M. and K.T. wrote the original draft; all authors reviewed the final version of the manuscript and agreed to its content before submission.

Competing interests. The authors declare no conflict of interest.

Funding. The long-term monitoring of Arctic charr in lava caves is funded by the Icelandic Research Fund, RANNÍS (research grant nos. 120227 and 162893). E.A.M. was supported by the Icelandic Research Fund, RANNÍS (grant no. 162893) and NERC research grant awarded to M.B.M. (grant no. NE/R011109/1). M.B.M. was supported by a University Research Fellowship from the Royal Society (London). C.A.L. and B.K.K. were supported by Hólar University, Iceland. The Titan Xp GPU used for this research was donated to K.T. by the NVIDIA Corporation.

Acknowledgements. We would like to acknowledge the many field assistants and colleagues who helped collect the fish data throughout the years. We are especially grateful to Katja Räsänen, Anett Relient, Mathias Lherbier, Kári H. Árnason and Árni Einarsson.

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
