## [Peer Review File · Royal Society Open Science]

Review History

RSOS-201768.R0 (Original submission)

Review form: Reviewer 1

Is the manuscript scientifically sound in its present form?

Yes

Are the interpretations and conclusions justified by the results?

Yes

Is the language acceptable?

Yes

Do you have any ethical concerns with this paper?

No

Have you any concerns about statistical analyses in this paper?

No

Recommendation?

Accept as is

Comments to the Author(s)

Well presented paper.

Review form: Reviewer 2

Is the manuscript scientifically sound in its present form?

Yes

Are the interpretations and conclusions justified by the results?

Yes

Is the language acceptable?

Yes

Do you have any ethical concerns with this paper?

No

Have you any concerns about statistical analyses in this paper?

No

Recommendation?

Accept with minor revision (please list in comments)

Comments to the Author(s)

This paper compared classical computer vision techniques with deep-learning techniques to compare performance for re-identification of individual Arctic charr through time. They found a U-net approach could reliably extract spots from photographs of Arctic charr and present two adapted match algorithms that can be applied in either a fully-automated (Groth matcher) or semi-automated (Growth matcher and astroalign algorithms) set-up. Generally, I enjoyed reading this paper and found the content interesting. It will be a very useful piece of literature for photo-identification programs, especially in cases where the volume of photographs is overwhelming the capacity of researchers to generate matches visually or through semi-automated software. Additionally, the authors clearly demonstrated the management applications of their pipeline via their discussion of the potential for false negatives or tag loss to underestimate survival rate from CJS models. I have a few minor edits to improve the readability of the paper, as the methods were hard to follow in some places and would benefit from some additional detail.

Line 102: 'the spots do not differ from any other spot, so improvement on current algorithms in this system will make these algorithms more widely applicable'. I am unsure what this means, could you please elaborate?

103 - This overview of the methods is useful but may be better suited to the methods section.

Line 129 (and throughout) - Perhaps referring to cave A as Cave A would make this easier to read.

Line 131- Some clarity would help here. Are these photos a subset from the full dataset taken from 2012-2017?

Line 132 – why are individuals only seen once or twice?

Lines 149 – 150 – the 41:11 split is confusing. Does this refer to the sample size in training and testing sets? Why do the sample sizes vary between the two sets?

161- is there a reference for this?

166- You could make this sentence clearer for the general readership of Royal Society Open Science. Perhaps a description of the parameters that you picked?

188 - 189 – Are these results more suited to the Results section?

191-197 and 198-2017 - Similarly, this seems suited to the Results and Discussion sections. The Methods section of this paper is long and a bit tedious to read so cutting results and discussion out of it will improve readability

Line 218: “are then removed’ should be ‘were removed’ – watch out for changes in tense throughout

Line 246 - what original paper that you are referring to? A reference here embedded in the text would be helpful.

Line 263- “This process is repeated until a good enough model is found’ - what qualifies as a good enough model? How are you judging the performance of the model?

Line 282: formatting of equation

Line 287: “This was so that” does not read nicely. Perhaps it could be “this was to ensure that”

Line 293: consider changing title of subheading, at the moment it reads as though you are presenting results. Perhaps “evaluating experimental results for spot matching”

Line 364- “An illustration of this is shown in Figure 11” – to save words you can just reference (Figure 11) at the end of the relevant sentence.

Line 413 – remove the comma after although

Line 426- consider replacing ‘good enough’ with sufficient or something less colloquial

Figures – all figures appear in greyscale but I can see from the figure legend that meant to be in colour – worth checking the correct versions are uploaded or whether they have been converted into greyscale post submission.

Figure 13 and 14 – would be clearer in colour

Review form: Reviewer 3

Is the manuscript scientifically sound in its present form?

Yes

Are the interpretations and conclusions justified by the results?

No

Is the language acceptable?

Yes

Do you have any ethical concerns with this paper?

No

Have you any concerns about statistical analyses in this paper?

No

Recommendation?

Major revision is needed (please make suggestions in comments)

Comments to the Author(s)

This paper proposes a solution to identify fishes from images using spot constellations, based on image processing tools and U-Net neural network.

This approaches seems to be interesting and crucial for the application.

To make this paper better, I propose to make a real conclusion and open some perspectives.

Here are some more detailed remarks:

- image processing, figure 6: you propose a threshold of 60. Can you precise the grey scale? Is the maximum at 256?
- line 161: can you detail the origins of the numbers in the formula ?
- I suggest to add a figure representing the U-Net architecture scheme;
- reference [33]: this reference is not explicit enough to be useful.

Decision letter (RSOS-201768.R0)

Dear Dr Mittell

On behalf of the Editors, we are pleased to inform you that your Manuscript RSOS-201768 "Re-identification of individuals from images using spot constellations; a case study in Arctic charr" has been accepted for publication in Royal Society Open Science subject to minor revision in accordance with the referees' reports. Please find the referees' comments along with any feedback from the Editors below my signature.

Please submit your revised manuscript and required files (see below) no later than 7 days from today's (ie 08-Jun-2021) date. Note: the ScholarOne system will 'lock' if submission of the revision is attempted 7 or more days after the deadline. If you do not think you will be able to meet this deadline please contact the editorial office immediately.

on behalf of Marta Kwiatkowska (Subject Editor)
openscience@royalsociety.org

Associate Editor Comments to Author:

Comments to the Author:

Please ensure that you carefully respond to and incorporate the suggested changes from reviewer 2 in particular (and also reviewer 3) in you revision - we will expect to a full point-by-point response to their queries/comments, especially if you opt not to include a tweak that has been suggested. Good luck and thanks in advance for your revision.

Reviewer comments to Author:

Reviewer: 1

Comments to the Author(s)

Well presented paper.

Reviewer: 2

Comments to the Author(s)

This paper compared classical computer vision techniques with deep-learning techniques to compare performance for re-identification of individual Arctic charr through time. They found a U-net approach could reliably extract spots from photographs of Arctic charr and present two adapted match algorithms that can be applied in either a fully-automated (Groth matcher) or semi-automated (Growth matcher and astroalign algorithms) set-up. Generally, I enjoyed reading this paper and found the content interesting. It will be a very useful piece of literature for photo-identification programs, especially in cases where the volume of photographs is overwhelming the capacity of researchers to generate matches visually or through semi-automated software. Additionally, the authors clearly demonstrated the management applications of their pipeline via their discussion of the potential for false negatives or tag loss to underestimate survival rate from CJS models. I have a few minor edits to improve the readability of the paper, as the methods were hard to follow in some places and would benefit from some additional detail.

Line 102: 'the spots do not differ from any other spot, so improvement on current algorithms in this system will make these algorithms more widely applicable'. I am unsure what this means, could you please elaborate?

103 - This overview of the methods is useful but may be better suited to the methods section.

Line 129 (and throughout) – Perhaps referring to cave A as Cave A would make this easier to read.

Line 131- Some clarity would help here. Are these photos a subset from the full dataset taken from 2012-2017?

Line 132 – why are individuals only seen once or twice?

Lines 149 – 150 – the 41:11 split is confusing. Does this refer to the sample size in training and testing sets? Why do the sample sizes vary between the two sets?

161- is there a reference for this?

166- You could make this sentence clearer for the general readership of Royal Society Open Science. Perhaps a description of the parameters that you picked?

188 - 189 – Are these results more suited to the Results section?

191-197 and 198-2017 - Similarly, this seems suited to the Results and Discussion sections. The Methods section of this paper is long and a bit tedious to read so cutting results and discussion out of it will improve readability

Line 218: “are then removed’ should be ‘were removed’ – watch out for changes in tense throughout

Line 246 - what original paper that you are referring to? A reference here embedded in the text would be helpful.

Line 263- “This process is repeated until a good enough model is found’ - what qualifies as a good enough model? How are you judging the performance of the model?

Line 282: formatting of equation

Line 287: “This was so that” does not read nicely. Perhaps it could be “this was to ensure that”

Line 293: consider changing title of subheading, at the moment it reads as though you are presenting results. Perhaps “evaluating experimental results for spot matching”

Line 364- “An illustration of this is shown in Figure 11” – to save words you can just reference (Figure 11) at the end of the relevant sentence.

Line 413 – remove the comma after although

Line 426- consider replacing ‘good enough’ with sufficient or something less colloquial

Figures – all figures appear in greyscale but I can see from the figure legend that meant to be in colour – worth checking the correct versions are uploaded or whether they have been converted into greyscale post submission.

Figure 13 and 14 – would be clearer in colour

Reviewer: 3

Comments to the Author(s)

This paper proposes a solution to identify fishes from images using spot constellations, based on image processing tools and U-Net neural network.

This approaches seems to be interesting and crucial for the application.

To make this paper better, I propose to make a real conclusion and open some perspectives.

Here are some more detailed remarks:

- image processing, figure 6: you propose a threshold of 60. Can you precise the grey scale? Is the maximum at 256?
- line 161: can you detail the origins of the numbers in the formula ?
- I suggest to add a figure representing the U-Net architecture scheme;
- reference [33]: this reference is not explicit enough to be useful.

===PREPARING YOUR MANUSCRIPT===

===PREPARING YOUR REVISION IN SCHOLARONE===

Author's Response to Decision Letter for (RSOS-201768.R0)

See Appendix A.

Decision letter (RSOS-201768.R1)

Dear Dr Mittell,

I am pleased to inform you that your manuscript entitled "Re-identification of individuals from images using spot constellations; a case study in Arctic charr" is now accepted for publication in Royal Society Open Science.

on behalf of Prof Marta Kwiatkowska (Subject Editor)
openscience@royalsociety.org

Appendix A

Manuscript ID: RSOS-201768

Response to reviewers

Title: "Re-identification of individuals from images using spot constellations; a case study in Arctic charr"

Dear Professor Marta Kwiatkowska,

We would like to thank you for reviewing our manuscript and giving us the opportunity to submit a revised version. We would also like to thank the reviewers for their comments. We have incorporated most of the suggestions made by the reviewers and have responded point-by-point below. Changes in the manuscript have been highlighted in red. We hope that the result of these revisions is satisfactory for publication in the "Life Sciences New Talent" collection, a special issue of Royal Society Open Science. Please note that reviewer 2 mentioned that their figures were in greyscale. Some of the figures that we have uploaded are in colour and we hope that this will be the case in the final version. Thank you again.

Response to reviewers

Reviewer 2

This paper compared classical computer vision techniques with deep-learning techniques to compare performance for re-identification of individual Arctic charr through time. They found a U-net approach could reliably extract spots from photographs of Arctic charr and present two adapted match algorithms that can be applied in either a fully-automated (Groth matcher) or semi-automated (Growth matcher and astroalign algorithms) set-up. Generally, I enjoyed reading this paper and found the content interesting. It will be a very useful piece of literature for photo-identification programs, especially in cases where the volume of photographs is overwhelming the capacity of researchers to generate matches visually or through semi-automated software. Additionally, the authors clearly demonstrated the management applications of their pipeline via their discussion of the potential for false negatives or tag loss to underestimate survival rate from CJS models. I have a few minor edits to improve the readability of the paper, as the methods were hard to follow in some places and would benefit from some additional detail.

Line 102: 'the spots do not differ from any other spot, so improvement on current algorithms in this system will make these algorithms more widely applicable'. I am unsure what this means, could you please elaborate?

To make this clearer this has been changed to: "In our case study on Arctic charr, the spots are generally consistent in size and colour, so improvement on current algorithms in this system with these generic spots should make these algorithms more widely applicable."

103 – This overview of the methods is useful but may be better suited to the methods section.

Moved to end of the methods overview. Line 141

Line 129 (and throughout) – Perhaps referring to cave A as Cave A would make this easier to read.

This has been changed.

Line 131- Some clarity would help here. Are these photos a subset from the full dataset taken from 2012-2017?

This has been changed to make it clearer that it is a separate dataset and not a subset of the first one that is described: "To independently verify the performance of the methods, we used a second dataset of 346 images also collected between 2012 and 2017, but from a different cave (hereafter

“Cave B”). This second dataset included 165 resolved recaptures of 160 known individuals (Figure 3b)”

Line 132 – why are individuals only seen once or twice?

It is the case that the vast majority of individuals are not recaptured many times and there is a skew towards fewer recaptures. Therefore, there are few confirmed recaptures from past studies that we can use to train or evaluate a model -- this is where the method presented here comes in. Changed to: “In these data, using previous methods of re-identification, most individuals are only seen once or twice.”

Lines 149 – 150 – the 41:11 split is confusing. Does this refer to the sample size in training and testing sets? Why do the sample sizes vary between the two sets?

We clarified this in the text. Baselines do not learn, we can use all 52 images to tune this. Learning methods need a hold-out set, we used an approximately 80:20 split.

161- is there a reference for this?

The adaptive mean thresholding algorithm is taken from OpenCV. The reference has been added and this is noted in the text. The link is

https://docs.opencv.org/3.4/d7/d1b/group_imgproc_misc.html#ga72b913f352e4a1b1b397736707afcde3.

It is an old and simple algorithm that is widely implemented.

166- You could make this sentence clearer for the general readership of Royal Society Open Science. Perhaps a description of the parameters that you picked?

We believe this has been clarified now. These are the parameters described in this section and we list them (125px, threshold of 80, etc.). Changed to: “This type of feature engineering approach requires domain knowledge and has many free parameters described above, such as sizes of Gaussian kernels and morphological operators, and threshold values.”

188 - 189 – Are these results more suited to the Results section?

191-197 and 198-207 - Similarly, this seems suited to the Results and Discussion sections. The Methods section of this paper is long and a bit tedious to read so cutting results and discussion out of it will improve readability

We included some early results throughout the methods that were used to build the pipeline and optimise the earlier algorithms as we believed that this makes it easier to follow. The spot detection and spot matching are two separate algorithms and the latter can only work if the former detects spots well. This means we thought that it may be useful to stop and have a look at spot detection performance before getting into later parts of the pipeline. However, to address these points we have moved a bulk of the evaluation of the spot detection to the results section and minimised this part of the methods. We hope that this helps with the readability of the manuscript.

Line 218: “are then removed” should be “were removed” – watch out for changes in tense throughout

Thank you. Changed and we have gone over the document again to try and catch any others that may have slipped through.

Line 246 - what original paper that you are referring to? A reference here embedded in the text would be helpful.

Added in.

Line 263- "This process is repeated until a good enough model is found' - what qualifies as a good enough model? How are you judging the performance of the model?

We have clarified this now: "This process is repeated until the proportion of inliers exceeds a threshold (as described later in this section), or it hits a set number of iterations."

Line 282: formatting of equation

The equation has formatted better thank you.

Line 287: "This was so that" does not read nicely. Perhaps it could be "this was to ensure that"

Changed.

Line 293: consider changing title of subheading, at the moment is reads as though you are presenting results. Perhaps "evaluating experimental results for spot matching"

Changed.

Line 364- "An illustration of this is shown in Figure 11" – to save words you can just reference (Figure 11) at the end of the relevant sentence.

Changed.

Line 413 – remove the comma after although

Changed.

Line 426- consider replacing 'good enough' with sufficient or something less colloquial

Changed.

Figures – all figures appear in greyscale but I can see from the figure legend that meant to be in colour – worth checking the correct versions are uploaded or whether they have been converted into greyscale post submission.

Thank you for highlighting this to us. This must have been converted post-submission. We will let the editorial office know.

Figure 13 and 14 – would be clearer in colour

As above.

Reviewer 3

This paper proposes a solution to identify fishes from images using spot constellations, based on image processing tools and U-Net neural network.

This approaches seems to be interesting and crucial for the application.

To make this paper better, I propose to make a real conclusion and open some perspectives.

Conclusion changed to: "Overall, we have demonstrated the feasibility of re-identifying individuals from photographs using these spot detection and matching algorithms. We have also shown how our re-identification method makes it possible to increase sample sizes without a great deal of extra effort in terms of data processing. This could be used to deal with different types of re-identification requirements. These results are promising and open up possibilities for re-identification from spots not only in these Arctic charr, but, in the longer term, in other species that exhibit spots. Finally, our results show that this tool can be used to support individual-based studies where tag-loss is an issue, by identifying its extent and reducing its effect."

Here are some more detailed remarks:

- image processing, figure 6: you propose a threshold of 60. Can you precise the grey scale? Is the maximum at 256?

Yes, all intensities are in the range of [0,255]. This has been added to the figure legend.

- line 161: can you detail the origins of the numbers in the formula?

This has been rephrased to make it clearer that it is an empirical equation we found on our data, not a general pre-existing formula: "We have determined that the following formula works well for our data based on empirical evaluation on the training images:"

- I suggest to add a figure representing the U-Net architecture scheme;

As the manuscript is already fairly long and contains quite a number of figures we haven't put in an additional figure here. We think that the figures specific to the workflow and results here are more relevant for the paper. However, we have specified where to find an overview figure for the U-Net architecture from the original paper, which we have cited: "is based on U-Net architecture [for example see Figure 1 in 30]".

- reference [33]: this reference is not explicit enough to be useful.

Thank you for pointing this out. This is a compilation error of the references. There should be a url for this and some other references. We have formatted correctly now.